# Leaching Kinetics of Aluminum from Alkali-Fused Spent Cathode Carbon Using Hydrochloric Acid and Sodium Fluoride

**Jie Yuan \*, Huijin Li and Shuang Ding**

School of Chemistry and Materials Engineering, Liupanshui Normal University, Liupanshui 553004, China; xihu1101@126.com (H.L.); xiaoxiangzi0927@163.com (S.D.)
\* Correspondence: yuanjieedu@163.com

**Abstract:** Abundant carbon resides in spent cathode carbon (SCC) of aluminum electrolysis and its high-purity carbon powder is conducive to high-value recycling. The alkali-fused SCC was separated and effectively purified using an HCl/NaF solution. Effects of particle size, leaching temperature, time, initial acid concentration, and sodium fluoride dosage, on the purity of carbon powder and aluminum removal rate, were investigated. Using aluminum as the research object, kinetics of aluminum acid leaching were examined by single-factor experiments. Results showed that under an initial 4 M HCl concentration, particle size D(50) = 67.49 μm, liquid-solid ratio of 15:1, 333 K, 120 min, 0.3 M NaF, carbon powder with ash level below 1% were obtained in subsequent purification of SCC. The leaching process was described by Avram equation, the model characteristic parameter was 0.75147 and the apparent activation energy was 22.056 kJ/mol, which indicated a mixed control mechanism between chemical reactivity and diffusion. The kinetic reaction equation of leaching aluminum from alkali-fused SCC in a mixed HCl/NaF system was established.

**Keywords:** spent cathode carbon of aluminum electrolysis; acid leaching; kinetics; Avrami equation





## 1. Introduction

Aluminum is an indispensable nonferrous metal critical for national economic development. Electrolysis of molten salt produces elemental aluminum via the Hall–Héroult process [1]. As the primary component of a reduction cell, carbon cathode used during aluminum electrolysis is prepared by anthracite, asphalt coke, natural and artificial graphite, roasted at high temperature. Cracks and corrosion pits form as the cathodic carbon block corrodes due to high-temperature liquid metal, molten alkaline electrolyte, and metallic sodium. High-temperature melt in the cell permeates the crevices and pits, which results in efficacy losses at the carbon cathode [2,3]. Additionally, secondary products (NaF, NaCN, $\beta$-$Al_2O_3$, etc.) occur via reactions between air and chemical species (sodium, aluminum, cryolite, etc.) attached to the cathode surface or crevices at high temperatures [4], which further aggravate carbon cathode failure. Generally, the electrolytic cell requires an overhaul and replacement of the cathode every 3–10 years [5]. Spent cathodic carbon is solid waste produced from electrolytic cell overhaul. In addition to large amounts of carbon [6](about 60~80 wt%), sodium fluoride (NaF), cryolite, and alumina removed during conventional alkali or acid leaching, there are some intractable non-carbon components such as complex aluminates and aluminosilicates in SCC from aluminum reduction cells. Non-carbon impurities primarily involve the high-temperature reaction products of alumina, cryolite, and silicon dioxide. These impurities make it difficult to improve the purity of carbon powder recovered from SCC, which is not conducive to high-value utilization of carbon powder [7].

Traditionally, comprehensive treatment of SCC from aluminum electrolysis focused on flotation [8], the pyrometallurgy process [9], and hydrometallurgical processes [10]. SCC is used as a fuel substitute in the cement [11] and nonferrous metallurgy industries [12]; however, despite utilizing the combustion characteristics of carbon, high-purity carbon

was not obtained. The thermal behavior of fluorides and cyanides in SCC was studied based on a TG/DSC-MS system by Li [13], who reported that fluorides were volatilized, and cyanides decomposed at high temperature, so non-hazardous treatment of SCC was conducted. Yao [14] purified SCC via NaOH-Na$_2$CO$_3$ binary molten salt roasting and water leaching and obtained a treated carbon purity of 96.98%. Leaching is a common hydrometallurgical method, and SCC treated by hydrothermal acid-leaching resulted in an aluminum leaching rate of ~80% under optimal conditions. The removal rates of impurities were not perfect, but the novel idea of silicon carbide preparation with hydrothermal acid leached coal gangue, and the waste cathode was reported [15].

A three-step process separated cryolite from SCC [16] via leaching with acidic anodizing wastewater, and a valuable, high-purity carbon (95.5%), was obtained. Lisbona [17] obtained carbon powder (purity ~95%) in various solutions. Defluorination of SCC was researched in an acidic iron-containing solution [18]. Low caustic leaching and liming (LCL&L) process developed by Rio Tinto Aluminum [19] is a typical method for SCC treatment. These conventional processes can remove some impurities from SCC; however, the carbon powder purities obtained were unsatisfying. Moreover, valuable carbonaceous materials were unrecoverable via pyrometallurgical techniques and new solid waste by-product residues were produced. Hydrometallurgy treatments are complex, produce low purity carbon powder, and novel, undesirable by-products. Recently, vacuum distillation, high-temperature graphitization, and others that improve carbon powder purity have been reported. For example, Xie [20] treated SCC in a joint controlling temperature-vacuum process, and optimized conditions produced a carbon purity of 97.89%. Wang [21] separated and recovered SCC by vacuum distillation. Cryolite, NaF, and sodium were separated effectively. At 1200 °C, the separation rate exceeded 80%, and the carbon powder purity > 91%. Patent CN108050848b [22] discloses a high-temperature continuous electric calcining furnace. It treats 2–50 mm of SCC particles at a feed rate of 500–600 kg/h to produce graphite with a fixed carbon content of >98%. Limited to high energy consumption and high-temperature resistant equipment, these technologies are difficult to apply in large-scale industrial treatment. Therefore, there is a strong incentive to investigate the deep purification process of spent cathode carbon.

In this study, SCC pretreated by sodium hydroxide alkali fusion was purified by a mixture of hydrochloric acid and NaF to remove acid-soluble inorganic impurities. Effects of initial acid concentration, reaction temperature, and NaF addition on the purification were studied. An acid leaching impurity and aluminum removal models were established based on a previous experimental procedure, and its apparent activation energy was calculated to clarify the leaching mechanism of impurities in a mixed acid solution.

## 2. Materials and Methods

### 2.1. Materials

SCC used in the work was discharged from an electrolytic aluminum plant in China. After crushing and grinding, the powder at different particle sizes was dried in an oven at 105 °C for 24 h to remove water. Sodium hydroxide, hydrochloric acid, and NaF were of analytical purity and obtained from Sinopharm Chemical Reagent Co., Ltd. (Beijing, China).

### 2.2. Procedure

Dried SCC, sodium hydroxide, and deionized water were mixed, stirred into a paste, and soaked for an additional 2 h. The mixture was loaded into a corundum crucible. After water evaporation, the powder was kept at 550 °C for 3 h under a nitrogen (99.99%) atmosphere in a muffle furnace (TF1200-100, Kejing Material Technology Co., Ltd.,Hefei, China, ~1200 °C). The sample was cooled, washed, and dried with deionized water until the filtrate was neutral. The final analysis and mineral content of the alkali-fusion treated powder are listed in Table 1 and Figures 1 and 2.

**Table 1.** Ultimate analysis of untreated and alkali-fused SCC/%.

| Element | | C | F | Na | Al | Ca | Si | O | Others |
|---|---|---|---|---|---|---|---|---|---|
| content | untreated | 61.06 | 14.37 | 8.71 | 7.09 | 1.35 | 0.43 | 5.47 | 1.52 |
| | treated | 93.26 | 0.74 | 0.48 | 1.46 | 2.38 | 0.57 | 0.73 | 0.38 |

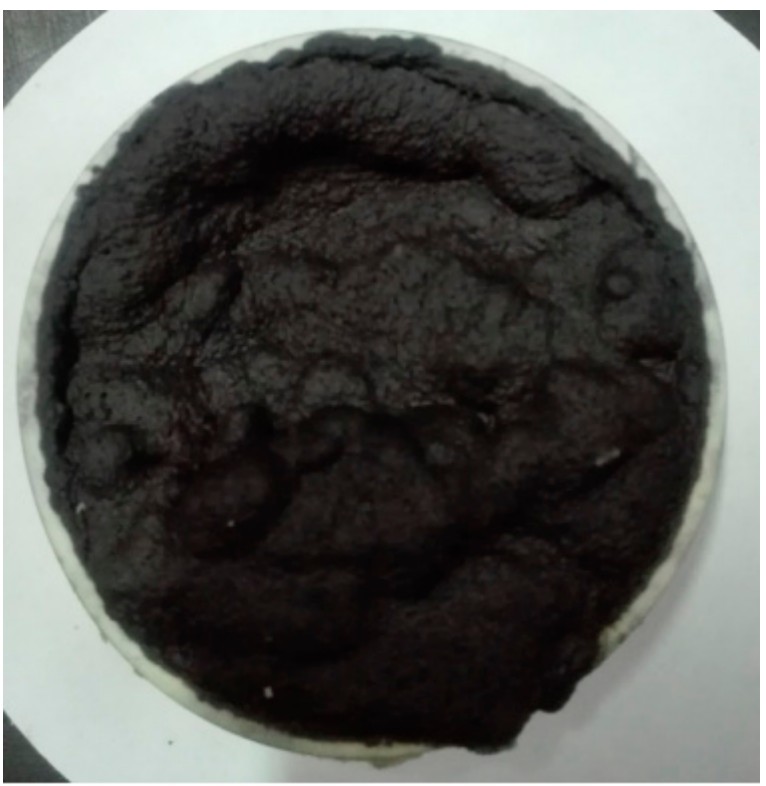

**Figure 1.** SCC after alkali fusion treatment.

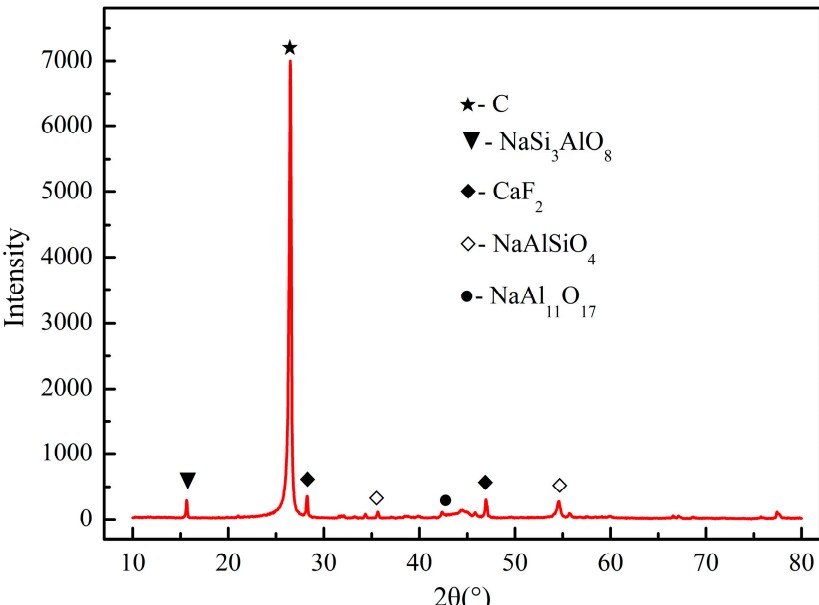

**Figure 2.** XRD pattern of alkali-fused SCC.

The alkali-fusion treated carbon powder was leached in a solution of hydrochloric acid and NaF. Leaching took place in a plastic beaker at constant temperature. Experimental

factors such as leaching temperature (303~333 K), time (~120 min), initial hydrochloric acid concentration (1~4 M), and NaF addition (0.1~0.4 M) on the ash levels of the leaching residue and aluminum leaching rate were investigated.

After alkali fusion and acid leaching, the powder was dried in a blast drying oven at 105 °C for 24 h, followed by ash content detection. To improve test efficiency and reduce experiment complexity, the ash content was expressed by the residual amount of purified carbon powder upon combustion at 800 °C in air for 4 h. Ash levels were calculated using Equation (1).

$$\eta_a = m_a/m_s \times 100\% \tag{1}$$

where $\eta_a$ is the ash content of purified SCC, (%) $m_a$ is the ash mass of purified SCC heated at 800 °C in the air for 4 h, g; $m_s$ is the mass of purified SCC combusted at 800 °C in air for 4 h, g.

The aluminum leaching rate was calculated using Equation (2).

$$\eta_A = (C_{A0} - C_A)/C_{A0} \times 100\% \tag{2}$$

where $\eta_A$ is aluminum leaching rate, %; $C_{A0}$ is the aluminum content in alkali fused SCC, %; $C_A$ is the aluminum level of SCC after alkali fusion and acid leaching, %.

### 2.3. Characterization

SCC was analyzed by X-ray fluorescence spectrometry (XRF). Carbon levels were determined by elemental analysis (EA, Vario E1 III, Elementar Company, Hanau, Germany) before XRF analysis. The samples with certain carbon, volatile matter, and moisture levels were burned in a muffle furnace at 800 °C for 4 h, followed by XRF analysis of the ash. Element levels were calculated from previous analysis results. Sample phases were detected by X-ray diffraction (XRD, Riguka D/max 2500 X-ray diffractometer, Japan Electronics Co., Ltd., Tokyo, Japan). Solid materials were broken into—200 mesh powder and dried at 105 °C for 12 h. Phase and microcrystalline structures were analyzed by XRD using a working voltage of 40 mV and a 100-mA current; Cu irradiation (Kα) was used at a step size of 10°/min from 10° to 80°. Scanning electron microscope analysis (SEM) and energy dispersive spectrometer analysis (EDS) were synchronous instruments in this study. A JSM-6360 LV scanning electron microscope made by Japan Electronics Co., Ltd. observed and analyzed the micro morphologies of the solid materials. EDS analysis was performed using an EDX-QENESIS 60S X-ray spectrometer (EDAX company, Philadelphia, PA, USA). A laser particle size analyzer (MS2000, Malvern, UK) was used for solid particle size analysis. Alcohol was the diffusing agent and the refractive index referenced to graphite.

### 3. Results and Discussion

#### 3.1. Characterization and Analysis

SCC subjected to alkali-fusion treatment is illustrated in Figure 1; the carbon powder was fluffy and expansive. Results of the complete analysis and phase characterization of carbon residue from alkali fused and washed SCC powder are given in Table 1 and Figure 2, respectively. Those results show the alkali fused residue contains some products soluble in strong acids and is consistent with initial exploratory experiments. From SCC purification results previously reported [23] and impurity reactivity in alkali fusion, there were three kinds of inorganic impurities in the alkali-fused carbon powder. The first type was unreacted hydroxide M(OH)x and oxide $M_2O_x$ (M was element Al, Fe, Mg, Ca, etc.) compounds. The second type included inorganics like calcium fluoride and complex aluminosilicates that do not react during alkali leaching. The last group includes products such as $Na_2SiO_3$, $SiO_2$, and $NaAlO_2$. Because they comprised less than 5 wt.%, inorganic impurities are not displayed in the XRD pattern accurately and comprehensively.

### 3.2. Thermodynamic Calculations

To improve the purity of carbon powder after alkali fusion treatment, a mixed solution of hydrochloric and hydrofluoric acids was used to leach the residue even further, according to types and properties of complex inorganics existing in alkali fused SCC powder. The possible reactions in acid leaching process are expressed in Equations (3)–(17). Figure 3 shows the relationship of $\Delta G$ vs. $T$ during acid leaching, as calculated using HSC-Chemistry 6.0 (Outotec).

$$Fe(OH)_3 + 3HCl = FeCl_3 + 3H_2O \tag{3}$$

$$Al(OH)_3 + 3HCl = AlCl_3 + 3H_2O \tag{4}$$

$$Mg(OH)_2 + 2HCl = MgCl_2 + 2H_2O \tag{5}$$

$$Ca(OH)_2 + 2HCl = CaCl_2 + 2H_2O \tag{6}$$

$$Fe_2O_3 + 6HCl = 2FeCl_3 + 3H_2O \tag{7}$$

$$Al_2O_3 + 6HCl = 2AlCl_3 + 3H_2O \tag{8}$$

$$MgO + 2HCl = MgCl_2 + H_2O \tag{9}$$

$$CaO + 2HCl = CaCl_2 + H_2O \tag{10}$$

$$CaF_2 + 2HCl = CaCl_2 + 2HF \tag{11}$$

$$NaAlSiO_4 + 4HCl = NaCl + AlCl_3 + SiO_2 + 2H_2O \tag{12}$$

$$SiO_2 + 6HF = H_2SiF_6 + 2H_2O \tag{13}$$

$$CaSiO_3 + 6HF + SiO_2 = CaSiF_6 + 2H_2O \tag{14}$$

$$H_2SiO_3 + 4HF = SiF_4(g) + H_2O \tag{15}$$

$$Na_2SiO_3 + 2HCl = 2NaCl + H_2SiO_3 \tag{16}$$

$$CaSiF_6 + HCl = H_2SiF_6 + CaCl_2 \tag{17}$$

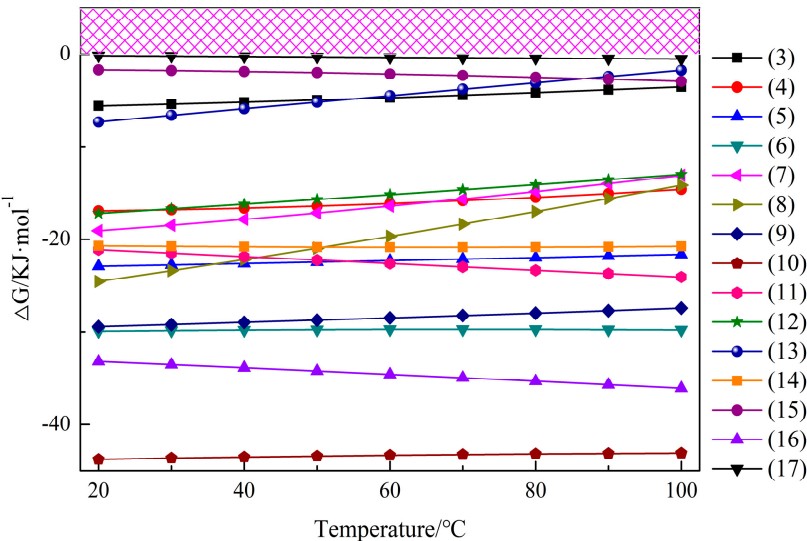

**Figure 3.** Relationship of functions' $\Delta G$ vs. $T$ in acid leaching process.

Thermodynamic calculation results in Figure 3 show the possible chemical reactions of inorganic impurities in alkali-fused carbon powder in hydrochloric acid and NaF mixed solution are thermodynamically feasible and reasonable. From 20−100 °C, acid leaching reactions are relatively simple; the products primarily contain water-soluble compounds or gaseous substances such as $SiF_4$. Thermodynamic analysis results provide theoretical

support for the effective removal of impurities in alkali-fused carbon powder in the mixed acid solution.

### 3.3. Effect of Experimental Factors on Leaching Efficiency

#### 3.3.1. Particle Size

Dried SCC powders with different particle sizes [−10~+60 mesh (D(50) = 304.36 μm), −60~+100 mesh (D(50) = 175.49 μm), −100~+200 mesh (D(50) = 94.56 μm), −200 mesh (D(50) = 55.51 μm)] were subjected to alkali fusion and mixed acid leaching for deep purification according to the experimental procedures outlined in Section 2.2. Experimental conditions of acid leaching were as follows: initial hydrochloric acid concentration of 4 M, liquid-solid ratio of 15:1 (constant), 333 K, and 0.3 M NaF. Effects of raw material particle size on carbon level in the purified powder and extraction rate of elemental aluminum during acid leaching were investigated. Figure 4 shows those experimental results.

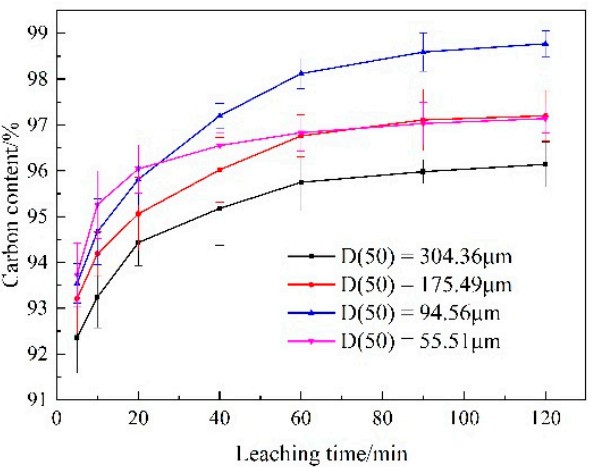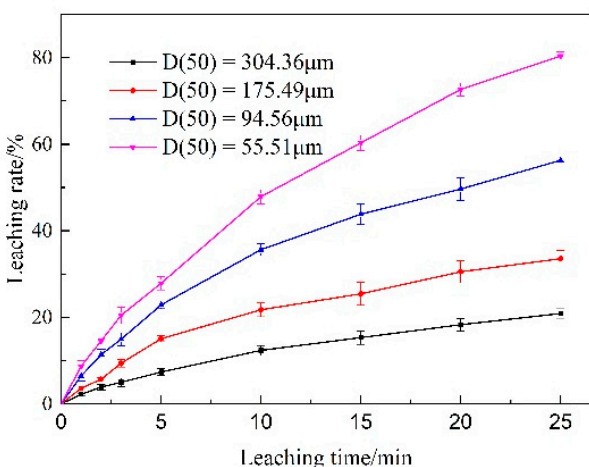

**Figure 4.** Effect of particle size on carbon content (**left**) and Al leching rate (**right**) (initial HCl concentration 4 mol/L, 333 K, NaF dosage 0.3 mol/L).

Sample particle size has a clear influence on the separation and leaching behavior of inorganic impurities. Smaller particle sizes correspond with faster reaction rates [24]. Higher impurity decomposition rates correspond to higher carbon powder purity and aluminum removal for leaching time less than 25 min. The highest purity was obtained from purification of −100 to +200 mesh (D(50) = 94.56 μm) raw materials at time above 60 min. The reason was that large particle size inorganic impurities were not exposed sufficiently, and impurities were readily trapped by carbon and lowered the acid reaction rate. When particle size was <200 mesh, according to the previous analysis [25], more impurities would disperse into the SCC powder. Therefore, considering experimental convenience and energy consumption, −100 mesh (D(50) = 67.49 μm) was the optimal particle size.

#### 3.3.2. Initial HCl Concentration

Figure 5 shows the effect of initial hydrochloric acid concentration on carbon powder purification after sodium hydroxide fusion treatment. Inorganic impurities in the alkali-fused SCC were effectively separated and removed in acid solution. The purity of carbon powder increased with the initial HCl concentrations, and the leaching rate of element aluminum also rose. At an initial concentration of 1 M, carbon powder purity increased from 92.41% after 5 min to 98.08% after 120 min, and aluminum leaching rate rose from 4.39% after 1 min to 49.61% after 25 min. At 4 M, carbon powder purity increased from 94.32% after 5 min to 98.81% after 120 min, and the aluminum leaching rate rose from 18.57% after 1 min to 81.06% after 25 min. Increasing the initial HCl concentration improved both carbon powder purity and aluminum leaching rate.

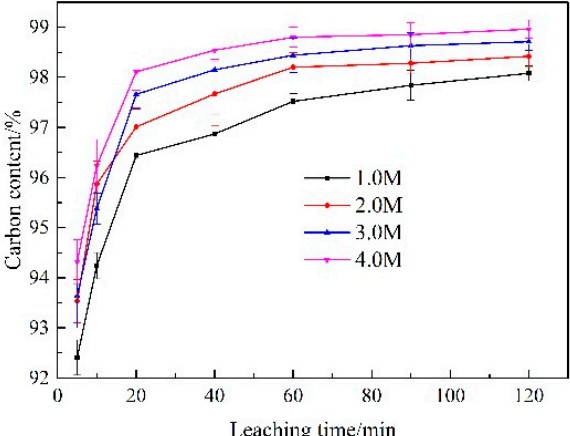
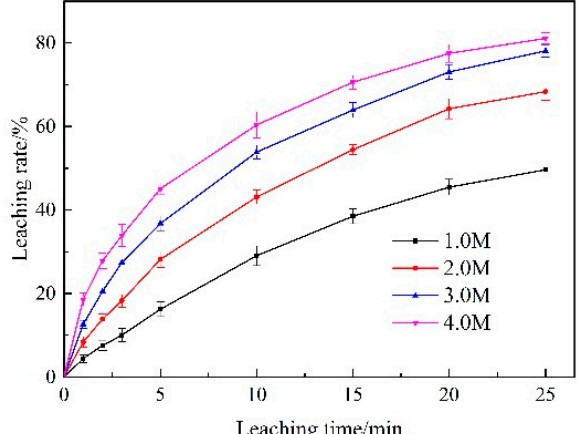

**Figure 5.** Effect of initial HCl concentration on carbon content (**left**) and Al leching rate (**right**) (particle size D(50) = 67.49 μm, 333 K, NaF 0.3 mol/L).

### 3.3.3. Temperature

Temperature played an important role in removing impurities by acid leaching [26]. The effects of acid leaching temperature on the carbon content of purified SCC and the aluminum removal rate were tested, and the results are illustrated in Figure 6. The purification efficiency of SCC increased with the gradual increase of reaction temperature. Carbon powders with purities > 98.9% and an aluminum leaching rate of 81.08% were obtained at temperatures above 333 K. Impurities in alkali fused and the washed carbon powder mainly included calcium fluoride and complex aluminosilicates. Calcium fluoride dissolves in acidic solutions, and aluminosilicate minerals react with acid to produce silicic acid and other substances. When temperature was above 323 k and leaching time was longer than 60 min, acid leaching temperature changes had little effect on impurity removal. There was an experimental phenomenon in which carbon content of the leaching residue decreased slightly at higher temperatures, primarily due to the increased volatility of hydrochloric acid caused by high temperatures. Volatilization by heating not only causes environmental pollution, but also reduces the acid concentration, which slightly impacts removal efficacy. Therefore, the optimal temperature of 60 °C was used.

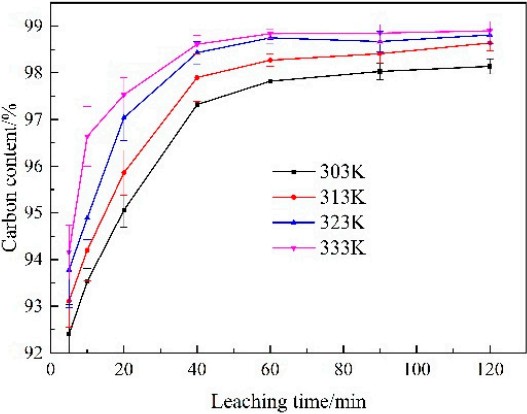
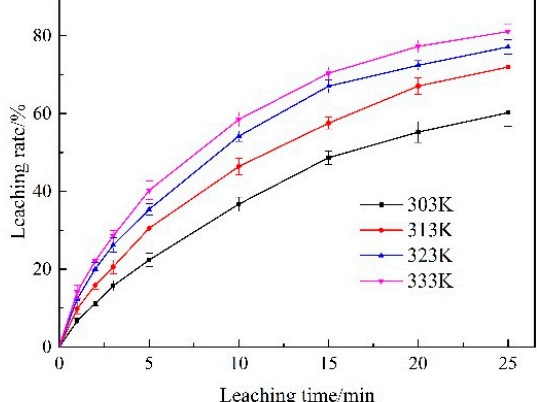

**Figure 6.** Effect of temperature on carbon content (**left**) and Al leching rate (**right**) (initial HCl concentration 4 mol/L, particle size D(50) = 67.49 μm, NaF 0.3 mol/L).

### 3.3.4. Sodium Fluoride Addition

As shown in Figure 7, carbon in the leaching residue increased significantly with added NaF in the acid leaching solution. The carbon levels increased in 20 min as NaF addition grew from 0.1–0.3 M, the corresponding purity increased from 97.12% to 97.99%. Increasing

the NaF dosage to 0.4 M increased the purity to 98.14% before trending downward. At an NaF concentration of 0.3 mol/L, the purity of purified carbon powder increased from 93.85% after 5 min to 98.92% after 120 min. NaF levels positively impacted the separation of inorganic impurities. With added NaF, hydrofluoric acid concentrations increased, which improved impurity removal.

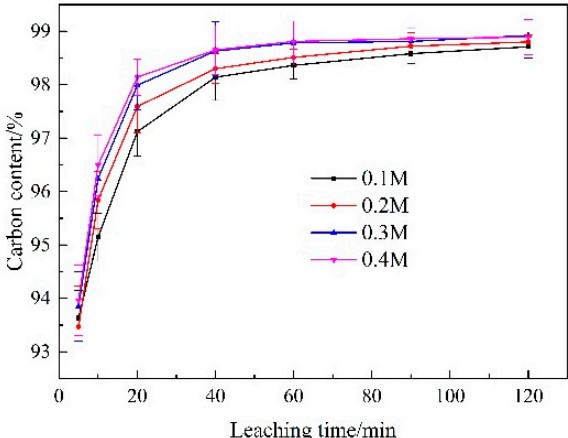 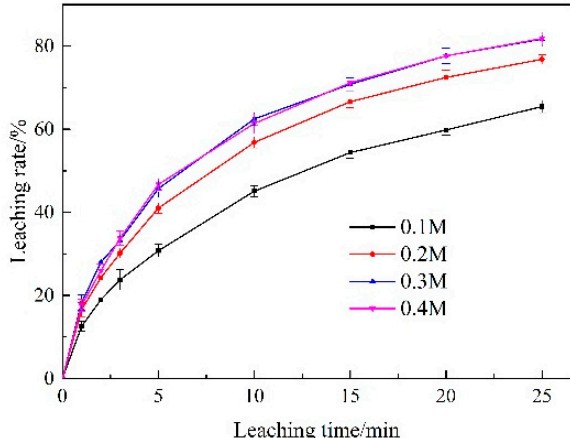

**Figure 7.** Effect of NaF addition on carbon content (**left**) and Al leching rate (**right**) (initial HCl concentration 4 mol/L, particle size D(50) = 67.49 μm, 333 K).

During acid leaching of fly ash, the leaching rate of aluminum improved significantly when using hydrochloric acid as a leaching agent and NaF as an additive. Fluoride ion in solution reacts with silicon in aluminum silicon glass to produce fluorinated silicon compounds that destroy the glass and improves aluminum reaction activity greatly. Reaction equations are as follows [27]:

$$H^+ + NaF = HF + Na^+ \tag{18}$$

$$Al_2O_3 \cdot SiO_2 + 4HF = SiF_4(g) + Al_2O_3 + 2H_2O \tag{19}$$

$$3Al_2O_3 \cdot SiO_2 + 6HF = H_2SiF_6 + 3Al_2O_3 + 2H_2O \tag{20}$$

$$H_2SiF_6 = SiF_6^{2-} + 2H^+ \tag{21}$$

$$Al_2O_3 + 6H^+ = 2Al^{3+} + 3H_2O \tag{22}$$

Reaction activities of some SCC impurities such as $SiO_2$, $NaAl_{11}O_{17}$, and $NaAlSiO_4$ are weak in hydrochloric acid systems, but solubilizing some of these impurities increases in hydrofluoric acid systems. Therefore, upon addition of NaF, carbon levels in the leaching residue showed a continuous upward trend (Figure 7). When NaF addition exceeded 0.3 M, residual impurities in SCC were not effectively removed in the existing leaching system; all impurities in the waste cathode were not removed completely in an HCl/HF solution, and the change of treated carbon powder purity was not obvious. In addition, NaF solubility in water is low and additional NaF in solution may complicate purified carbon powder washing. Solid carbon powder provided crystal nucleation sites, and porous carbon adsorbed on those crystalline particles. So, the optimal amount of NaF was 0.3 M.

### 3.3.5. Purified SCC Analysis

High purity carbon powder with ash levels below 1% was obtained by acid leaching the alkali-fused SCC in a hydrochloric acid/sodium fluoride solution. Figure 8 shows the XRD analysis of SCC and carbon powder after acid leaching, and Figure 9 shows the SEM-EDS results. After treatment, the ash content of purified carbon powder was less than 1%.

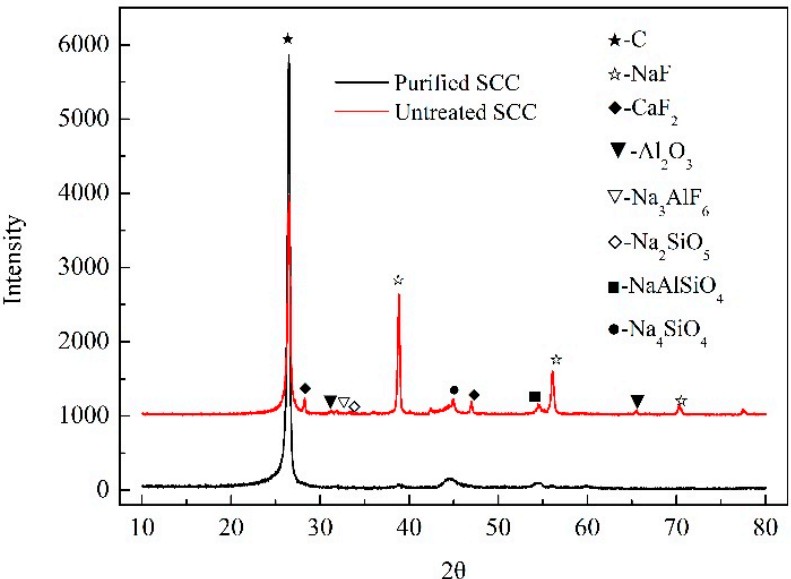

**Figure 8.** XRD patterns of SCC.

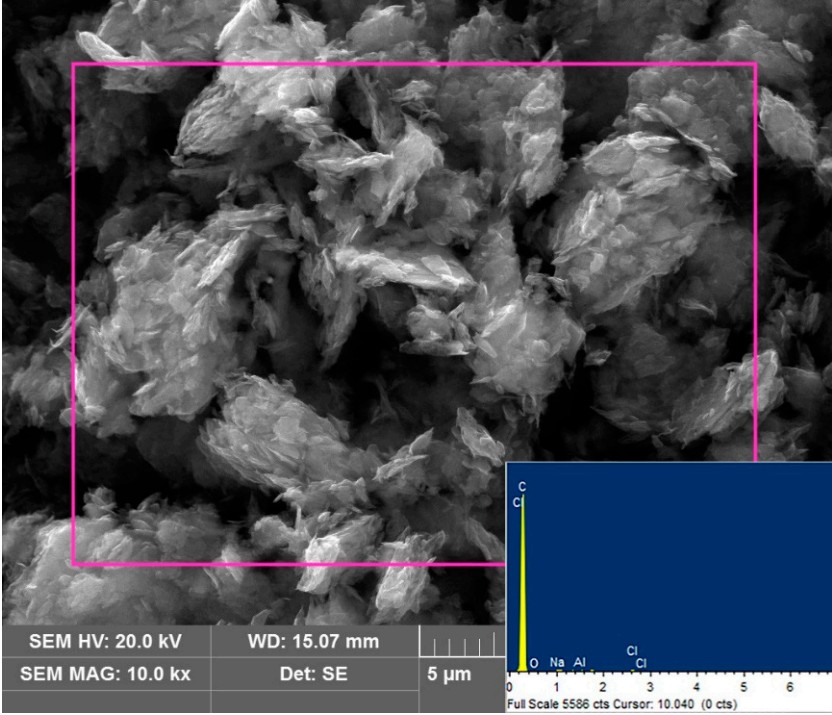

**Figure 9.** SEM-EDS images of SCC after alkali fusion-acid leaching treatment.

*3.4. Leaching Kinetic Analysis*

3.4.1. Kinetic Model

Aluminum, a primary element of SCC, was selected to explore the extraction efficiency during leaching; the leaching reaction mechanism was explained kinetically.

The Avrami Equation [28] was first used in the kinetics of nucleus growth in heterogeneous chemical reactions; it now uses in the leaching of many metals and metal oxides [29]. Equation (23) shows this equation:

$$-\ln(1 - \eta) = kt^n \tag{23}$$

where k is the apparent reaction rate constant; η is leaching rate, %; t is leaching time, min; n is reaction characteristic parameter.

The reaction characteristic parameter depends on mineral grain property and geometry, which reflects the leaching reaction mechanism. It relates only to the properties and geometry of solid grains and remains unchanged with reaction conditions [30]. As n < 1, the initial reaction rate is very high and decreases with leaching progress. As n > 1, the initial reaction rate approaches 0.

Taking the natural logarithm on both sides of Equation (23) yields the following Equation (24):

$$\ln[-\ln(1-\eta)] = \ln k + n \ln t \qquad (24)$$

### 3.4.2. Determination of Leaching Model Parameters

Results in Figure 4 (right) were substituted into Equation (24), and variations of $\ln[-\ln(1-\eta)]$ with ln t were plotted for the particle sizes as shown in Figure 10. The good linear correlation in Figure 10 indicated the Avrami model effectively describes the aluminum extraction from alkali-fused cathode carbon under different particle sizes. The linear fitting regression equations and correlation coefficient values are listed in Table 2.

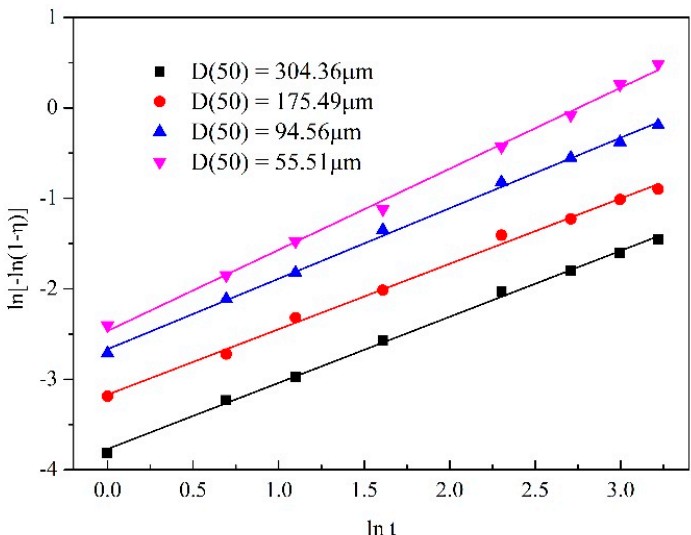

**Figure 10.** Plots of $\ln[-\ln(1-\eta)]$ vs. ln t in alkali fused SCC leaching process under different particle sizes.

**Table 2.** Fitting results between $\ln[-\ln(1-\eta)]$ and ln t under different particle size.

| Particle Size | Regression Equation | $R^2$ |
|---|---|---|
| D(50) = 304.36 μm | $\ln[-\ln(1-\eta)] = 0.73058\ln t - 3.76902$ | 0.99799 |
| D(50) = 175.49 μm | $\ln[-\ln(1-\eta)] = 0.72284\ln t - 3.16824$ | 0.99552 |
| D(50) = 94.56 μm | $\ln[-\ln(1-\eta)] = 0.77899\ln t - 2.66602$ | 0.99755 |
| D(50) = 55.51 μm | $\ln[-\ln(1-\eta)] = 0.89611\ln t - 2.46394$ | 0.99666 |

The apparent reaction rate constant, k, relates to solution concentration, mineral particle size, and temperature. Combined with the Arrhenius equation, k is expressed as shown in Equation (25).

$$k = k_0 \, C_{HCl}^a \, C_{NaF}^b \, D^d \exp(-Ea/RT) \qquad (25)$$

where $k_0$ is the frequency factor; $C_{HCl}$ is the initial concentration of hydrochloric acid, M; $C_{NaF}$ is NaF addition, M; D is the particle size, μm; Ea is the activation energy, J/mol; T is temperature, K; R is the universal gas constant, 8.314 J/(K·mol); a is the hydrochloric acid concentration reaction order; b is the NaF concentration reaction order; d is the particle size influence index.

Taking logarithms of Equation (25) yielded Equation (26).

$$\ln k = \ln k_0 + a\ln C_{HCl} + b\ln C_{NaF} + d\ln D - Ea/RT \tag{26}$$

Based on Equation (26), others were fixed conditions and variable particle size, variations of lnk with lnD were plotted in Figure 11. The slope of the straight line was −0.82962, which was the particle size influence index.

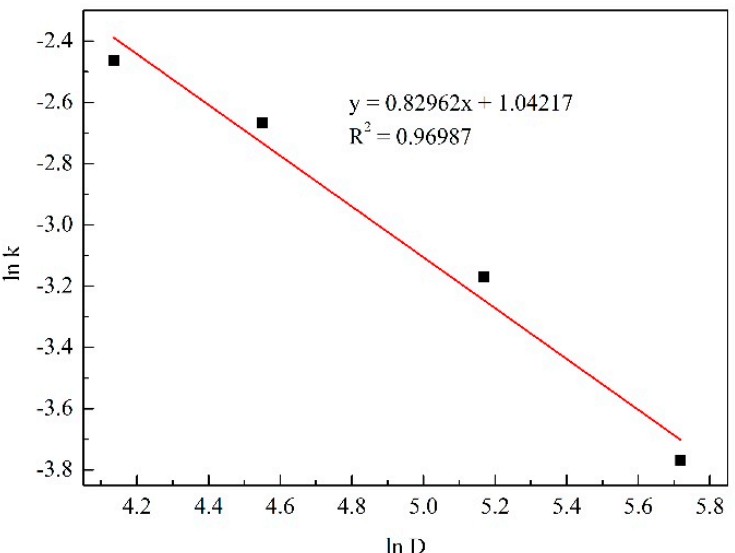

**Figure 11.** Plot of lnk vs. lnD in alkali fused SCC leaching process.

Results shown in Figure 5(right) were substituted into Equation (24), and the variations of ln[−ln(1 − η)] with lnt were plotted for initial acid concentrations in Figure 12. The linear fitting regression equations and correlation coefficients are listed in Table 3. The straight lines in Figure 12 and fitting curve correlation coefficients $R^2$ in Table 3 showed good linear correlations between ln[−ln(1 − η)] and lnt under different initial acid concentrations.

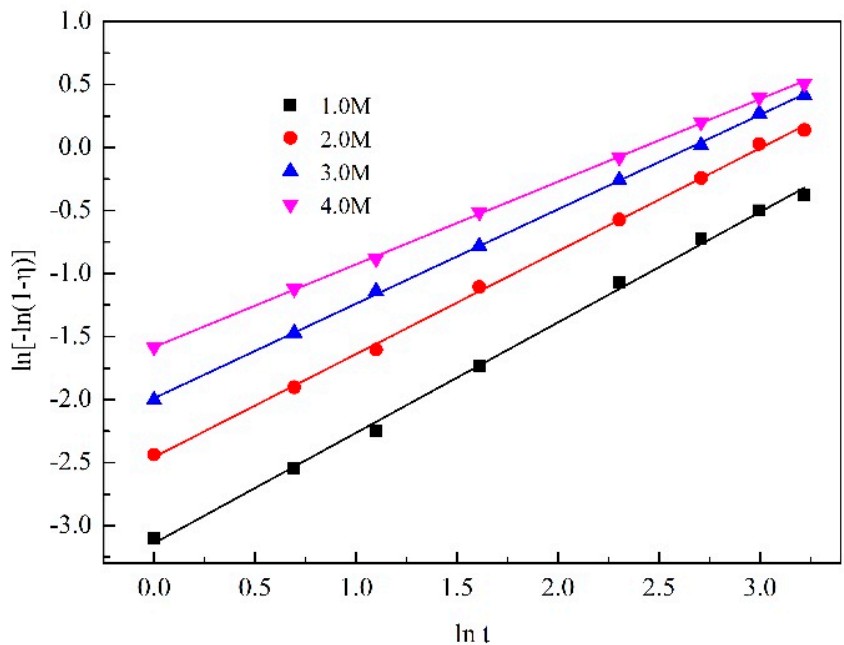

**Figure 12.** Plots of ln[−ln(1 − η)] vs. lnt in alkali fused SCC leaching process under different acid concentrations.

**Table 3.** Fitting results between $\ln[-\ln(1 - \eta)]$ and $\ln t$ under different acid concentrations.

| Initial Acid Concentration | Regression Equation | $R^2$ |
|---|---|---|
| 1 mol/L | y = 0.87611x − 3.13885 | 0.99762 |
| 2 mol/L | y = 0.81689x − 2.45518 | 0.99885 |
| 3 mol/L | y = 0.74951x − 1.98818 | 0.99964 |
| 4 mol/L | y = 0.65595x − 1.58231 | 0.99957 |

Varying the initial acid concentration while fixing the other variables gave variations of $\ln k$ with $\ln C_{HCl}$ and are plotted in Figure 13. The slope of the straight line was 1.10841, which was the hydrochloric acid concentration influence index.

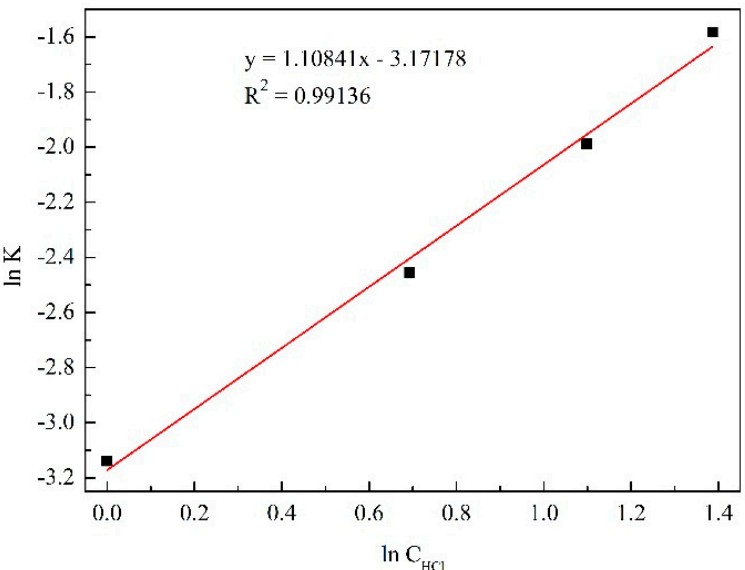

**Figure 13.** Plot of $\ln k$ vs. $\ln C_{HCl}$ in alkali fused SCC leaching process.

Results shown in Figure 6(right) were substituted into Equation (24), and variations of $\ln[-\ln(1 - \eta)]$ with $\ln t$ were plotted as a function of leaching temperature (Figure 14). The linear fitting regression equations and correlation coefficients are listed in Table 4.

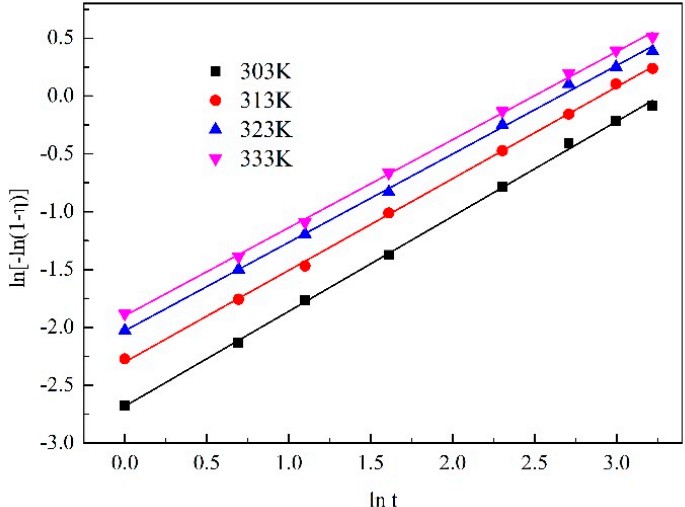

**Figure 14.** Plots of $\ln[-\ln(1 - \eta)]$ vs. $\ln t$ in alkali fused SCC leaching process under different temperatures.

**Table 4.** Fitting results between $\ln[-\ln(1 - \eta)]$ and $\ln t$ under different temperatures.

| Temperature | Regression Equation | $R^2$ |
|---|---|---|
| 303 K | y = 0.82056x − 2.67984 | 0.99899 |
| 313 K | y = 0.79246x − 2.29784 | 0.9993 |
| 323 K | y = 0.76404x − 2.02802 | 0.99847 |
| 333 K | y = 0.7609x − 1.89813 | 0.99900 |

Varying temperature and keeping other variables fixed yielded the variations of $\ln k$ with $1/T$, which were plotted and shown in Figure 15. The slope of the straight line was 1.10841, which was hydrochloric acid concentration influence index. The slope was $-E_a/R$, and an apparent reaction activation energy $E_a$ 22.056 kJ/mol was obtained. In general, diffusion controls most metallurgical processes when the apparent activation energy is less than 10 kJ/mol; chemical reactivity primarily controls processes with apparent activation energies > 40 kJ/mol. However, there is mixed control when the apparent activation energy ranges between 10–40 kJ/mol. Therefore, the leaching process of alkali-fused SCC in an HCl/NaF system was controlled by a mixed mechanism of chemical reactivity and diffusion.

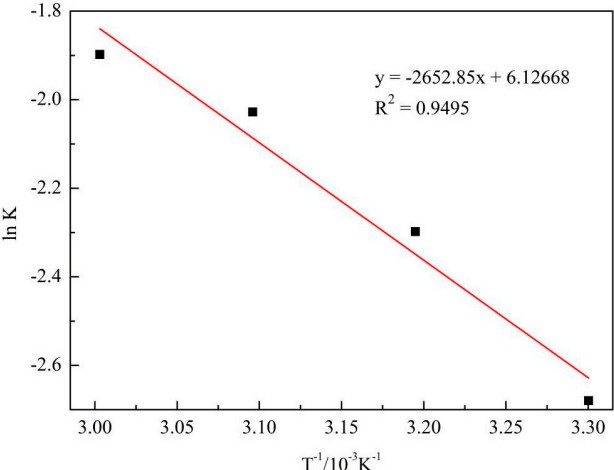

**Figure 15.** Plot of $\ln k$ vs. $T^{-1}$ in alkali fused SCC leaching process.

Results shown in Figure 7 (right) were substituted into Equation (24) to yield the fitting results given in Figure 16 and Table 5.

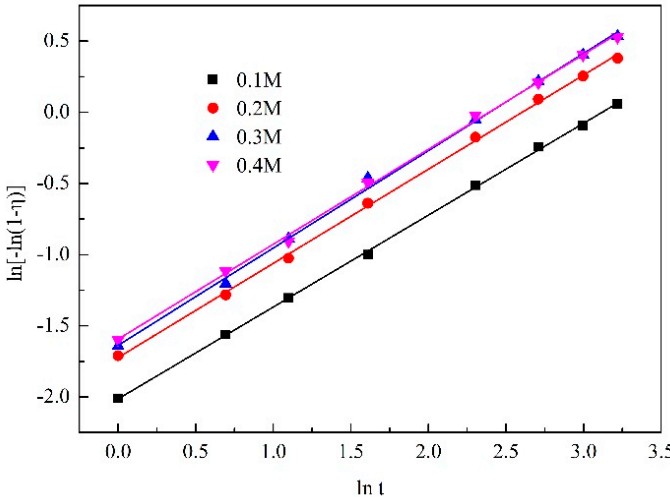

**Figure 16.** Plots of $\ln[-\ln(1 - \eta)]$ vs. $\ln t$ in alkali fused SCC leaching process under different NaF additions.

**Table 5.** Fitting results between $\ln[-\ln(1-\eta)]$ and $\ln t$ under different NaF additions.

| NaF Concentration | Regression Equation | $R^2$ |
|---|---|---|
| 0.1 mol/L | y = 0.64577x − 2.01349 | 0.99951 |
| 0.2 mol/L | y = 0.66183x − 1.72284 | 0.99899 |
| 0.3 mol/L | y = 0.68402x − 1.63707 | 0.99782 |
| 0.4 mol/L | y = 0.66698x − 1.59361 | 0.99858 |

NaF addition varied, and others were fixed; the variations of $\ln k$ with $\ln C_{NaF}$ are plotted in Figure 17. The NaF addition influence index was 0.30758.

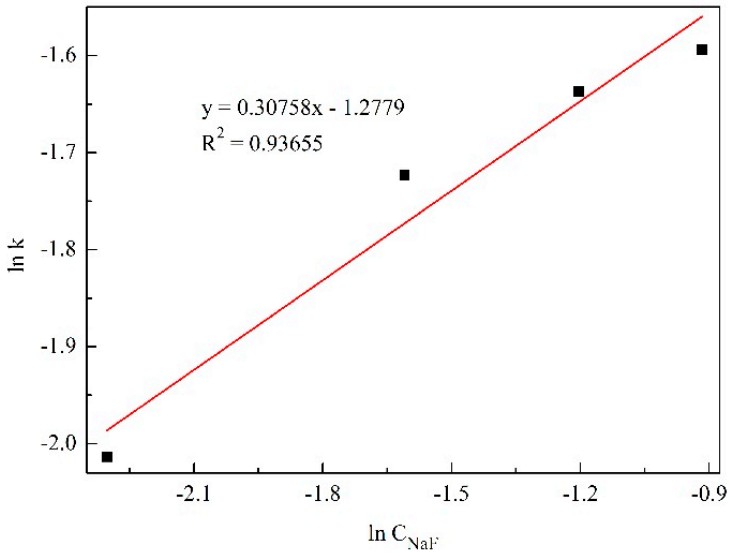

**Figure 17.** Plot of $\ln k$ vs. $\ln C_{NaF}$ in alkali fused SCC leaching process.

### 3.4.3. Determination of the Kinetic Equation

Tables 2–5 show a good linear relationship between $\ln[-\ln(1-\eta)]$ and $\ln t$. According to the data fitted by all Avrami equations, the average reaction characteristic parameter $n$ value was 0.75147. Based on Equations (23) and (25), variations of $-\ln(1-\eta)$ with $C_{HCl}^{1.10841} C_{NaF}^{0.30758} D^{-0.82962} \exp\left(-2652.85/T\right) t^{0.75147}$ were plotted in Figure 18. The slope of the straight line obtained by fitting was the frequency factor, $k_0$ (2.78788).

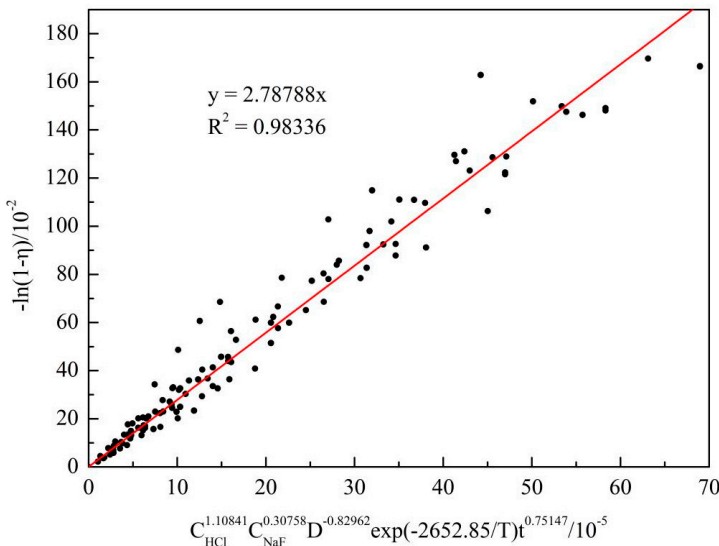

**Figure 18.** Plot of $-\ln(1-\eta)$ vs. $C_{HCl}^{1.10841} C_{NaF}^{0.30758} D^{-0.82962} \exp(-2652.85/T) \, t^{0.75147}$.

Equation (27) gives the kinetic equation of aluminum extraction from alkali-fused SCC from aluminum smelters by leaching in an HC/NaF mixture.

$$-\ln(1-\eta) \;=\; 2.78788 C_{HCl}^{1.10841} C_{NaF}^{0.30758} D^{-0.82962} \exp(-2652.85/T) t^{0.75147} \tag{27}$$

## 4. Conclusions

Spent cathode carbon of aluminum electrolysis was treated with alkali fusion and then purified deeply in an HCl/NaF solution. Useful results have been achieved.

(1)   Alkali-fused SCC powders from aluminum electrolysis were purified in a mixed solution of hydrochloric acid and sodium fluoride; the residual ash content was <1%.

(2)   Leaching of alkali fusion treated SCC powder in a hydrochloric acid and sodium fluoride system is described by the Avrami equation. The reaction characteristic parameter was 0.75147 and the apparent activation energy was 22.056 kJ/mol. Leaching was controlled by a mixed mechanism of chemical reaction and diffusion.

(3)   Kinetic equation of aluminum extraction was: $-\ln(1-\eta) = 2.78788 C_{HCl}^{1.10841} C_{NaF}^{0.30758} D^{-0.82962} \exp(-2652.85/T) t^{0.75147}$.

**Author Contributions:** Conceptualization, J.Y.; methodology, J.Y.; software, H.L. and S.D.; formal analysis, H.L.; investigation, J.Y. and H.L.; resources, J.Y.; data curation, H.L.; writing-original draft preparation, J.Y. and S.D.; writing-review and editing, S.D.; visualization, J.Y.; supervision, S.D.; project administration, J.Y.; funding acquisition, J.Y. All authors have read and agreed to the published version of the manuscript.

**Funding:** This research was supported by National Natural Science Foundation of China (51904150), Basic Research Program of Guizhou Province ([2020]1Y225), Guizhou Province Ordinary Universities Scientific Talents Project (KY [2019]056), Liupanshui Key Laboratory of Aluminum Production and Application (52020-2019-05-09), Liupanshui Normal University Scientific Research Foundation (LPSSYKYJJ201904).

**Conflicts of Interest:** The authors declare that there is no conflict of interest regarding the publication of this paper.

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
