# Peer review of "Leaching Kinetics of Aluminum from Alkali-Fused Spent Cathode Carbon Using Hydrochloric Acid and Sodium Fluoride"

_processes, doi:10.3390/pr10050849_

Round 1
Reviewer 1 Report
In this paper, the leaching process of leaching alkali-fused spent cathode carbon using hydrochloric acid and sodium fluoride was studied, the optimized parameters were obtained. The leaching kinetics of aluminum was then investigated and the kinetic reaction equation was established. However, I don’t recommend it to be published for the following reasons.
(1) The alkaline-acidic process for leaching spent cathode carbon is not innovative. A similar method has been reported in 2012(Trans. Nonferrous Met. Soc. China 22(2012) 222−227).
(2) The authors have not referred the mechanism of the process(alkali-fused, then leaching by hydrochloric acid and sodium fluoride).
(3) The readers have not known the composition of the spent cathode carbon the authors used for alkali-fused. Furthermore, one cannot get the precise results of the element content through XRF and cannot identify one phrase by only one characteristic peak in XRD pattern.
(4) There are some syntax errors.
Author Response
Thank you for your comments. These comments are all valuable and very helpful for revising and improving our paper, as well as the important guiding significance to our research.
(1) The alkaline-acidic process for leaching spent cathode carbon is not innovative. A similar method has been reported in 2012(Trans. Nonferrous Met. Soc. China 22(2012) 222−227).
Reply: We are grateful for the comments regarding our paper.
The paper published in 2012(Trans. Nonferrous Met. Soc. China 22(2012) 222−227) was a two-step alkaline-acidic leaching process, the carbon purity was 96.4%. This report was referred in our previous work (Industrial & Engineering Chemistry Research, 2018, 57(22):7700-7710.). And Shi is a very familiar and respected teacher of mine.
In our work we treated spent cathode carbon in alkali fusion and acid leaching process. The difference of our work and Shi’ work are alkali fusion vs alkali leaching, mixed solution of hydrochloric acid and sodium fluoride vs hydrochloric acid, carbon purity of above 99% vs 96.4.
(1) In sodium hydroxide alkali fusion, a series of phase transitions of inorganic impurities in spent cathode carbon which are difficult to occur in basic solution at room temperature are initiated, new acid soluble products are formed and separated from carbon.
(2) Purification ability of mixed solution of hydrochloric acid and sodium fluoride is higher than that of hydrochloric acid.
(3) NaF is inorganic impurity in spent cathode carbon, it is useful for comprehensive utilization of spent cathode carbon.
(4) Therefore, carbon purity got in our work was higher.
Above all, we believe our work is valuable
(2) The authors have not referred the mechanism of the process(alkali-fused, then leaching by hydrochloric acid and sodium fluoride).
Reply: Thanks for your valuable question.
In this work, the research was focused on leaching process of alkali-fused spent cathode carbon by hydrochloric acid and sodium fluoride.
The possible chemical reactions (eq.3~eq.25) of complex inorganic impurities in molten alkali were predicted and verified by thermodynamic calculations in our previous work.
The “previous work” is not published and we will contribute it to the journal Journal of material cycles and waste management in this week.
Na3AlF6 + 4NaOH = NaAlO2 + 6NaF + 2H2O |
(3) |
6NaF + 3SiO2 + 2Al2O3 = 3NaAlSiO4 + Na3AlF6 |
(4) |
6NaF + 9SiO2 + 2Al2O3 = 3NaAlSi3O8 + Na3AlF6 |
(5) |
2NaOH + 2SiO2 + Al2O3 = 2NaAlSiO4 + H2O |
(6) |
NaAlSi3O8 = NaAlSiO4 + 2SiO2 |
(7) |
NaAlSi3O8 + 4NaOH = NaAlSiO4 + 2Na2SiO3 + 2H2O |
(8) |
3Al2O3·2SiO2 + 4SiO2 + 6NaOH = 6NaAlSiO4 + 3H2O |
(9) |
Na2O·2SiO2 + Al2O3 = 2NaAlSiO4 |
(10) |
Na2O·SiO2 + SiO2 + Al2O3 = 2NaAlSiO4 |
(11) |
2Na2O·SiO2 + 3SiO2 + 2Al2O3 = 4NaAlSiO4 |
(12) |
NaAlO2 + SiO2 = NaAlSiO4 |
(13) |
2NaAlSi3O8 = Na2O·2SiO2 + Al2O3·SiO2 + 3SiO2 |
(14) |
2NaAlSi3O8 = Na2O·2SiO2 + Al2O3·2SiO2 + 2SiO2 |
(15) |
Al2Si2O7 + 2NaOH = 2NaAlSiO4 + H2O |
(16) |
Al2Si2O7 + 6NaOH = 2NaAlO2 + 2Na2SiO3 + 3H2O |
(17) |
Al2Si2O7 + 4NaOH = 2NaAlO2 + Na2Si2O5 + 2H2O |
(18) |
3Al2O3·2SiO2 + 6NaOH + 4SiO2 = 6NaAlSiO4 + 3H2O |
(19) |
Al2O3·2SiO2·2H2O + 2NaOH = 2NaAlSiO4 + 3H2O |
(20) |
Al2O3·2SiO2 + 2NaOH = 2NaAlSiO4 + H2O |
(21) |
SiO2 + NaOH = Na2SiO3 + H2O |
(22) |
Al2SiO5 + 2NaOH + SiO2 = 2NaAlSiO4 + H2O |
(23) |
Al2SiO5 + 2NaOH + 5SiO2 = 2NaAlSi3O8 + H2O |
(24) |
Al2O3 + 2NaOH = 2NaAlO2 + H2O |
(25) |
Fig.1 Relationship between functions’ ΔG and T in alkali-fusion process
(3) The readers have not known the composition of the spent cathode carbon the authors used for alkali-fused. Furthermore, one cannot get the precise results of the element content through XRF and cannot identify one phrase by only one characteristic peak in XRD pattern.
Reply: Thank you very much for pointing this out.
Table 1 is ultimate analysis of SCC. carbon is the main element in SCC, and its content is about 61%. The other elements in SCC are Na, F, Al, Si, O, and some other trace elements.
Table 1 Ultimate analysis of SCC
Element |
C |
F |
Na |
Al |
O |
Si |
Ca |
K |
Fe |
Others |
Content/% |
61.06 |
14.37 |
8.71 |
7.09 |
5.47 |
0.43 |
1.35 |
0.68 |
0.43 |
0.29 |
In this work, we tried to leaching alkali-fused SCC in mixed solution of hydrochloric acid and sodium fluoride. Therefore, alkali-fused SCC was the raw material in this research. According to your suggestion, we give the composition of the spent cathode carbon in P4 Table 1.
In the future experiments, we will choose appropriate methods for more accurate element analysis.
XRF is one of the commonly elemental analysis instruments, XRD is a commonly phase analysis method. In this study, ultimate analysis was carried out by XRF and phase analysis was by XRD. These are common analysis in many papers. In the whole work, ash content of purified carbon powder is calculated by the residual amount of purified carbon powder upon combustion at 800 °C in air for 4 h, and the content of Al is analyzed by XRF. Because aluminum content accounts for a high proportion in spent cathode carbon, Al content measured by XRF can be analyzed and summarized as the experimental results.
(4) There are some syntax errors.
Reply: Thank you very much for pointing this out.
MJEditor (www.mjeditor.com) provided English editing services during the preparation of this manuscript.
We have checked our paper carefully according to your suggestion. We believe that the paper is much better after incorporating the comments.
Finally, thanks for your patience and thoughtful review. With your comments the manuscript has become more reasonable and we have learnt a lot from your thorough thinking and painstaking work.

Reviewer 2 Report
The in favour points of this paper:
- Experiments were carried out in a systematically way to investigate the leaching of alkali-fused SCC.
- Both thermodynamics and kinetics of the leaching process were discussed.
My comments on this paper are listed below, please answer carefully:
- Page 3, line 98, the format of Eq. (1) needed to be aligned properly;
- Page 3, line 128, it's better to present some data on the chemical composition of the SCC (for example, the carbon%) before the alkaline fusion to show to support your argument if there is any.
- For Fig. 3, it may be better to change some of the lines to different types of dashed lines so that it is easier to identify the individual reaction.
- It is admitted that the standard Gibbs free energy values for the reactions listed in Fig. 3 are helpful in some way. But what may be more important is the real Gibbs free energy values (ΔG0 vs ΔG). So, in addition to the ΔG0 plot, it is better to have an equilibrium calculation of the alkaline fused system at given initial conditions and calculate the final equilibrium condition to determine thermodynamically the possible phases or species at the end.
- Page 6, when investigating the particle size effect, have the authors studied the change of the morphologies of the particles with different sizes before and after the alkali-fusion process?
- Is each of the data points in Fig. 4 the average of repeated experiments, if so, how many repetitions were performed?
- Page 6, line 190, please explain what does it mean by saying dispersed into the SCC powder? if there are any SEM images, it will be better to show them here.
- Page 7, line 218, the authors believed that the reactions were not sensitive to temperature variations. Least, Fig. 6 demonstrated quite a significant change in the reaction kinetics. Did you mean that above 303K, the final carbon content of the samples was almost the same for the reactions at different temperatures?
- Please note that the SiF4 is in a gaseous state.
- Page 9, what are the criteria for the SCC after treatment to be reusable in terms of the carbon content?
- Page 10, have the authors tried other models except for the Avrami model for the potential kinetic parameters determination?
- Page 3, line 105, "and acid leached SCC" -> "SCC after acid leaching"; Page 6, line 197, "higher"->"the"; Page 8, line 230, "rose"->"increased";
Author Response
Thank you for your comments. These comments are all valuable and very helpful for revising and improving our paper, as well as the important guiding significance to our research.
(1) Page 3, line 98, the format of Eq. (1) needed to be aligned properly;
Reply: Thank you very much for pointing this out. We feel really sorry for our mistake. The format of Eq. (1) has been aligned properly.
(2) Page 3, line 128, it's better to present some data on the chemical composition of the SCC (for example, the carbon%) before the alkaline fusion to show to support your argument if there is any.
Reply: Thank you very much for your suggestion.
Table 1 is ultimate analysis of SCC. carbon is the main element in SCC, and its content is about 61%. The other elements in SCC are Na, F, Al, Si, O, and some other trace elements.
Table 1 Ultimate analysis of SCC
Element |
C |
F |
Na |
Al |
O |
Si |
Ca |
K |
Fe |
Others |
Content/% |
61.06 |
14.37 |
8.71 |
7.09 |
5.47 |
0.43 |
1.35 |
0.68 |
0.43 |
0.29 |
In this work, we tried to leaching alkali-fused SCC in mixed solution of hydrochloric acid and sodium fluoride. Therefore, alkali-fused SCC was the raw material in this research. According to your suggestion, we give the composition of the spent cathode carbon in P4 Table 1.
(3) For Fig. 3, it may be better to change some of the lines to different types of dashed lines so that it is easier to identify the individual reaction.
Reply: Thank you very much for your suggestion.
According to your suggestion, we tried to change some of the lines to different types of dashed lines, however, the dashed lines cannot be clearly displayed. Maybe identify the individual reaction by line color is acceptable.
(4) It is admitted that the standard Gibbs free energy values for the reactions listed in Fig. 3 are helpful in some way. But what may be more important is the real Gibbs free energy values (ΔG0 vs ΔG). So, in addition to the ΔG0 plot, it is better to have an equilibrium calculation of the alkaline fused system at given initial conditions and calculate the final equilibrium condition to determine thermodynamically the possible phases or species at the end.
Reply: This suggestion is one of the most helpful comments on our work from three reviewers for revising and improving our paper, as well as the important guiding significance to our research. In the future research we will pay much more attention to equilibrium calculation.
In this work, the total content of inorganic matter in the alkali-fused carbon powder was less than 7 wt%, which was only suitable for qualitatively analyzing the change rule of non-carbon impurities in spent cathode carbon during alkali fusion. All of the impurities cannot be identified by characteristic peaks in XRD pattern. Calculation of the final equilibrium condition has some difficulties, and the results may be not accuracy.
In the future, the real Gibbs free energy values will be key point of thermodynamic calculation in similar studies.
(5) Page 6, when investigating the particle size effect, have the authors studied the change of the morphologies of the particles with different sizes before and after the alkali-fusion process?
Reply: Thank you very much for pointing this out.
Particle sizes of spent cathode carbon and alkali-fused spent cathode carbon are different. In this work, the research was focused on leaching process of alkali-fused spent cathode carbon by hydrochloric acid and sodium fluoride. Therefore, alkali-fused SCC was the raw material and we studied the change of the morphologies of the particles with different sizes before and after the alkali-fusion process.
This is a good research point and we will pay attention to your suggestion in the future research.
(6) Is each of the data points in Fig. 4 the average of repeated experiments, if so, how many repetitions were performed?
Reply: some of data points in Figs in the work was the average of repeated experiments. When there were abnormal points on the curve, we repeated the experiment, and the value was average of three repeated experiments.
(7) Page 6, line 190, please explain what does it mean by saying dispersed into the SCC powder? if there are any SEM images, it will be better to show them here.
Reply: Thanks for your comment.
Fig.12 is carbon content of SCC in different sizes in our previous work(Industrial & Engineering Chemistry Research, 2018, 57(22):7700-7710.)
Carbon materials in SCC consist of graphite and amorphous carbon. Graphite has good lubricity and ductility, and low hardness. Impurities in sample with poor ductility are harder than graphite, therefore, impurities can be grounded to smaller sized power more easily. So, carbon content of sample decreases with decreased powder particle size, which has good agreement with the conclusions of Li (Adv. Mater. Res.2014, 881−883, 1660−1664) and D.F. Lisbona (Ind. Eng. Chem. Res. 2012, 51 (39), 12712−12722 ).
(8) Page 7, line 218, the authors believed that the reactions were not sensitive to temperature variations. Least, Fig. 6 demonstrated quite a significant change in the reaction kinetics. Did you mean that above 303K, the final carbon content of the samples was almost the same for the reactions at different temperatures?
Reply: Thank you very much for pointing this out. We feel really sorry for our mistake.
In this paragraph, P7, line211, we pointed “Temperature played an important role in removing impurities by acid leaching”. Line218, the right expression is” When temperature was above 323k and leaching time was longer than 60min, acid leaching temperature changes had little effect on impurity removal.”.
We sincerely apologize for this mistake.
(9) Please note that the SiF4 is in a gaseous state.
Reply: Thanks for your comment.
SiF4 is a gaseous state. We have corrected this carelessness in the revised draft, page 8 eq.(19).
(10) Page 9, what are the criteria for the SCC after treatment to be reusable in terms of the carbon content?
Reply: Thanks for your comment.
After purification, the ash content of carbon powder was less than 1%. Graphite is the main carbon in purified powder. It can be used to prepare cathode, high-temperature crucibles, graphite molds, graphite bearings and graphite electrodes for smelting. Carbon with much impurities may cause cathode in aluminum electrolysis decreased conductivity. Therefore, carbon powder with high purity can be used in more ways.
(11) Page 10, have the authors tried other models except for the Avrami model for the potential kinetic parameters determination?
Reply: Thanks for your valuable question.
Based on the diversity of aluminum compounds in spent cathode carbon, we choose Avrami model for the potential kinetic parameters determination after referring to some literature (Miner Process Extr Metallurgy Review, 2018, 41: 1-10; Chinese J. Nonfer Metal, 2018,28(1):175-182). Unreacted shrinking core model is also a suitable model and we will try the model.
(12) Page 3, line 105, "and acid leached SCC" -> "SCC after acid leaching"; Page 6, line 197, "higher"->"the"; Page 8, line 230, "rose"->"increased";
Reply: Thank you for your suggestion.
We have checked and revised our paper carefully according to your suggestion.
Finally, thanks for your patience and thoughtful review. With your comments the manuscript has become more reasonable and we have learnt a lot from your thorough thinking and painstaking work.

Reviewer 3 Report
The authors have conducted a comprehensive study about the effects of leaching temperature, time, initial hydrochloric acid concentration, and NaF addition on the purification of SCC. The purpose and necessity of this study are fully defined. The composition and structural development of materials were characterized by the methods of XRD, SEM/EDS, and XRF.
This paper is in good shape but some comments should be answered by the authors before the publication of this paper:
1) P2L83: please cite the reference for “previous work”;
2) P3L91-93: please list a table for these experimental factors, such as the temperature range, concentration range, and so on;
3) P9, Figure 8: please draw the XRD pattern of SCC before alkali fusion so that readers can compare them;
4) P9, Figure 9: please arrange the SEM and EDS images.
Author Response
Thank you for your comments. These comments are all valuable and very helpful for revising and improving our paper, as well as the important guiding significance to our research.
1) P2L83: please cite the reference for “previous work”;
Reply: Thank you very much for pointing this out.
The “previous work” is not published and we will contribute it to the journal Journal of material cycles and waste management in this week.
Fig.1 flow chart of SCC purification
The first step was impregnation. SCC powder was mixed with sodium hydroxide in an appropriate proportion. Distilled water and surface modifier absolute alcohol were added, and the mixture was stirred evenly for 3h. Liquid-solid ratio of distilled water and SCC in the impregnation process was 3 mL/g, and the liquid-solid ratio of absolute alcohol and the powder was 0.1 mL/g.
The second step was evaporation. The above slurry was heated to be reduced to a pulpy consistency.
The third step was alkali fusion. The viscous mixture placed in corundum crucibles was heated in a tubular furnace (TF1200-100, Kejing Material Technology Co., Ltd, China, ~1200℃). Alkali fusion process was performed in a nitrogen (99.99%) atmosphere.
The fourth step was washing and acid leaching. Alkali-fused SCC was washed in distilled water several times until water pH reached to 7. The filtrate was collected for the impregnation step. Washing residue was leached in a mixed solution of hydrochloric acid (initial concentration of 4 mol/L) and sodium fluoride (dosage of 0.3 mol/L). Acid-leaching residue was dried to determine carbon purity. The washing and acid leaching process was implemented using a magnetic stirrer (DF101-S, Kezuo Technology Co., Ltd., China) at a constant temperature of 60 ℃ for 2 h.
2) P3L91-93: please list a table for these experimental factors, such as the temperature range, concentration range, and so on.
Reply: Thanks for your valuable suggestion.
We have rived in the revised manuscript.
“temperature (303~333K), time (~120min), initial hydrochloric acid concentration (1~4M), and NaF addition (0.1~ 0.4M) on the ash levels of the leaching residue…”
3) P9, Figure 8: please draw the XRD pattern of SCC before alkali fusion so that readers can compare them.
Reply: Thanks for your valuable suggestion.
We have rived in the revised manuscript.
4) P9, Figure 9: please arrange the SEM and EDS images.
Reply: Thanks for your valuable suggestion.
We have rived in the revised manuscript.
Finally, thanks for your patience and thoughtful review. With your comments the manuscript has become more reasonable and we have learnt a lot from your thorough thinking and painstaking work.
